# A paired-end whole-genome sequencing approach enables comprehensive characterization of transgene integration in rice

Wenting Xu[1], Hanwen Zhang[1], Yuchen Zhang[1], Ping Shen[2], Xiang Li[3], Rong Li[1] & Litao Yang [1✉]

Efficient, accurate molecular characterization of genetically modified (GM) organisms is challenging, especially for those transgenic events transferred with genes/elements of recipient species. Herein, we decipher the comprehensive molecular characterization of one novel GM rice event G281 which was transferred with native promoters and an RNA interference (RNAi) expression cassette using paired-end whole genome sequencing (PE-WGS) and modified TranSeq approach. Our results show that transgenes integrate at rice chromosome 3 locus 16,439,674 included a 36 bp deletion of rice genomic DNA, and the whole integration contains two copies of the complete transfer DNA (T-DNA) in a head-to-head arrangement. No unintended insertion or backbone sequence of the transformed plasmid is observed at the whole genome level. Molecular characterization of the G281 event will assist risk assessment and application for a commercial license. In addition, we speculate that our approach could be further used for identifying the transgene integration of cisgenesis/intragenesis crops since both ends of T-DNA in G281 rice were from native gene or elements which is similar with that of cisgenesis/intragenesis. Our results from the in silico mimicking cisgenesis event confirm that the mimic rice *Gt1* gene insertion and its flanking sequences are successfully identified, demonstrating the applicability of PE-WGS for molecular characterization of cisgenesis/intragenesis crops.

[1] National Center for the Molecular Characterization of Genetically Modified Organisms, Joint International Research Laboratory of Metabolic and Developmental Sciences, School of Life Sciences and Biotechnology, Shanghai Jiao Tong University, Shanghai 200240, China. [2] Development Center of Science and Technology, Ministry of Agriculture of People's Republic of China, Beijing 100025, China. [3] Technical Center for Animal, Plant and Food Inspection and Quarantine of Shanghai Customs, Shanghai 200135, China. ✉email: yylltt@sjtu.edu.cn

The commercialization of genetically modified organisms (GMO) has brought great benefits to farmers, consumers, and the environment[1]. At the request of all countries and regions, the biosafety of each GMO event should be thoroughly evaluated in terms of molecular characterization, food safety, and environmental safety. Only events that satisfy risk assessment should be approved for production and commercialization[2].

Molecular characterization of GMOs at the genome level often generates information on exogenous DNA integration and its inheritable stability, and a full understanding of exogenous DNA integration is a fundamental step in molecular characterization[3]. Exogenous DNA integration in GMOs includes integrated exogenous genes, integration sites, flanking DNA sequences of all inserts, the copy number of all inserts, and the presence or absence of unexpected vector backbone sequences[4]. This analysis plays a pivotal role in GMO biosafety supervision, quantitative detection, establishment of detection methods, and analysis of the content and traceability of GMO products[5]. This information is crucial for labeling for assessing the phenotype structural stability, food safety, and commercial value of new GM materials[6,7]. In the guidelines for GMO risk assessment, several techniques have been specified for probing exogenous DNA integration including PCR-based chromosome walking, Real-time PCR and Southern blotting, etc.[5]. PCR-based chromosome walking approaches coupled with Sanger sequencing are used for identifying exogenous DNA integration and flanking recipient genomic regions, but these techniques also have shortcomings, especially sequence bias and failure to generate comprehensive data, which can lead to missing insertion sites, especially for GM events with multiple exogenous DNA insertions[8]. Real-time PCR and Southern blotting are recommended for determining the copy number of exogenous genes and unintended insertion of backbone sequences of transformed vectors[5]. However, real-time PCR is easy to acquire ambiguous results when studying genes with multiple copies and/or tandem repeats of exogenous genes[8]. Southern blotting requires a large number of samples, takes a long time, requires technically accomplished researchers, and the selection of different enzyme digestion sites can lead to differences in copy number[5]. Molecular characterization of most commercialized GM crop events in the past few decades has employed the above methodologies, but publically available data is not always complete and accurate due to the inherent limitations of the methodologies[5].

High-throughput next-generation resequencing (NGS) is increasingly used in many research areas as the cost decreases dramatically. NGS has been successfully coupled with bioinformatics pipelines for full or partial molecular characterization of GMO events based on the resequencing technique. Zhang et al., identified integration sites and their flanking DNA sequences in transgenic cattle[9]. Two transgenic glyphosate-tolerant soybean insertion sites and flanking sequences were identified using whole-genome sequencing (WGS)[10]. The precise insertion loci and copy number of transfer DNAs (T-DNAs) in transgenic rice plant lines SNU-Bt9–5, SNU-Bt9–30, and SNU-Bt9–109 were determined using NGS-based molecular characterization methods, and the data were comparable to those obtained from Southern blotting analysis[11]. Furthermore, corresponding bioinformatics pipelines for analysis of NGS data have been developed[5,12,13]. NGS technology combined with target enrichment was used to identify genetic integrations from complex food/feed samples[14–16]. Previous studies have successfully employed WGS coupled with paired-end sequencing (PE-WGS) to identify transgene insertion sites and their flanking sequences. However, molecular characterization using current PE-WGS still remains challenging, especially for transgene insertions with endogenous recipient DNA or DNA sequences that are highly similar to host species[8,17].

As recombinant techniques develop and more gene functions are clearly annotated, novel GM crops are being produced via cisgenesis and intragenesis, along with traditional transgenesis introduction of exogenous genes[18]. Cisgenesis involves donor DNA fragments from the species itself or from a cross-compatible species without vector backbone DNA except T-DNA border sequences[19]. Intragenesis is more similar to cisgenesis, but intragenesis allows the creation of novel combinations of DNA fragments, such as combining functional genetic elements like promoters and terminators, and antisense or RNA interference (RNAi) cassettes[19]. Although cisgenesis and intragenesis mostly involve their own DNA fragments, a risk assessment should still be performed, including molecular characterization[7,20]. However, reports on molecular characterization of cisgenesis and intragenesis in crops are limited, and there is only one report on deciphering the T-DNA insertion of partial cisgenic lines of apple 'Gala' using inverse PCR[17]. The molecular characterization analysis of the cisgenesis and intragenesis using NGS tools was not reported, since picking up a cis gene from an endogenous genomic background is challenging.

In order to explore the potential of PE-WGS approach in molecular characterization of transgenesis, cisgenesis, and intragenesis, one transgenic rice event and one in silico mimicking cisgenesis event were used as examples. We performed molecular characterization of these two events using PE-WGS and a dedicated pipeline and discussed the characterization of cisgene integrations in cisgenesis/intragenesis.

## Results

**Sequencing data from PE-WGS.** Transgenic rice G281 (GM) and non-GM line Xiushui 110 rice were sequenced, and 11.7 GB and 12.2 GB data comprising 54,130,964 and 54,819,968 pairs of 100 bp reads were acquired, respectively (NCBI access number: SRR18236702 and SRX3923908). General sequencing coverage was up to 28.91× for GM rice and 29.28× for WT rice. Additionally, both ends of the 53,545,140 and 54,233,374 read pairs were mapped to the rice reference genome (Table 1). The calibration weight R with values of 0.986 for both GM and WT was used to modify bias among the mapped sequencing data, which was calculated from the percentage of read pairs that mapped to the reference genome among the total number of trimmed paired reads.

**Exogenous DNA integration sites and flanking sequences were successfully identified and confirmed.** All trimmed read pairs were analyzed according to the pipelines depicted in Fig. 1. For the G281 line, 53,545,140 read pairs and 2604 read pairs mapped to the rice reference genome and the transformed plasmid, respectively. A total of 332 read pairs were extracted for which one end mapped to the rice reference genome and the other mapped to the transformed plasmid using the original TranSeq pipeline[5]. Using the modified TranSeq pipeline, 253 read pairs from native genome sequence were discarded and only 69 read pairs were retained as candidate reads for determining the T-DNA insertion site and its flanking sequences (Table 1). A comparison of the extracted candidate reads from these two pipelines were shown in Table 2. Among the 253 filtered out read pairs, 17 pairs mapped to Chr 01, 73 mapped to Chr 02, eight pairs mapped to Chr 03, and 155 pairs mapped to Chr 10. All the sequences of these read pairs were from the homologous sequences between transformation vectors and rice genome, such as Gt1 promoter and *CYP81A6* gene, etc. We also found this kind

**Table 1 Sequencing data estimates and mapping results analysis.**

| Sample | Total trimmed read pairs | Sequencing depth (D) | Read pairs mapped to host genome | Coverage calibrator (R) | Read pairs mapped to plasmid | Reads clustered around insertion regions | |
|---|---|---|---|---|---|---|---|
| | | | | | | Not filtered | Filtered |
| Xiushui (WT) | 54819968 | 29.28 | 54233374 | 0.99 | 332 | 262 | 5 |
| G281(GM) | 54130964 | 28.91 | 53545140 | 0.99 | 2604 | 322 | 69 |

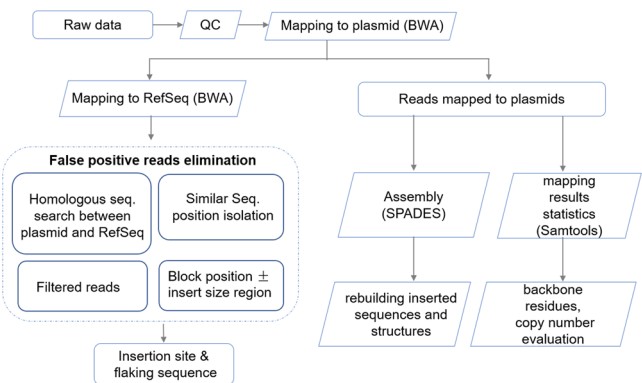

**Fig. 1 Modified bioinformatics pipeline for molecular characterization of transgenic plants using the PE-WGS approach.** QC means quality control to obtain the clean sequencing data from raw data, includes steps of removing adapters, low-quality sequences, and contaminated sequences. The Dotted box shows the procedure for eliminating the reads from native genome sequence.

of reads in WT line as well. Among the 69 potential candidate reads, 62 read pairs mapped to Chr 03, two pairs mapped to Chr 02, and one pair mapped to Chr 04, 06, 08, 11, and 12. After considering the sequencing depth (28.91×) and BLASTN analysis, the 62 read pairs mapped to Chr 03 were confirmed as true candidates (Table 2). The other seven sporadic read pairs mapped to other chromosomes were also confirmed to be untrue according to BLASTN analysis, IGV view, and PCR amplification. BLASTN analysis showed that the distance or locations between paired ends were contradictory among these seven pairs (Supplementary Table 1). The PCR amplification results of these seven pairs showed completely same amplifications (No amplicon or same amplicons) between G281 and WT line (Supplementary Fig. 1). IGV view of these seven sporadic read pairs also showed that all seven pairs can not be visualized within reasonable distance or locations according to rice reference genome (Supplementary Fig. 2). In whole genome re-sequence analysis, the sporadic read is hard to avoid because of the limitation of re-sequencing technique, we could use different methods to exclude these reads, such as BLASTN and IGV analysis, and PCR amplification.

The 69 candidate read pairs were mapped to rice reference genome sequence (GCA_001433935.1) by clustering pattern analysis, and the IGV visualization showed that only one transgene integration was introduced at locus 16,439,674 of Chr 03 (Fig. 2a). The assembled eight contigs (Supplementary Sequence File 1) generated from the 62 read pairs in de novo analysis clearly showed the T-DNA integration site at Chr 03 locus 16,439,674 with a 36 bp rice host genome DNA deletion, and two contigs with lengths of 478 bp and 434 bp comprising the 5′ and 3′ end flanking sequences adjacent to transgene insertion (Fig. 2b). Further conventional PCR results showed that the

expected DNA fragments covering the adjacent regions of the inserted site were only observed in G281 rice, while no amplicons were observed in WT rice. Sanger sequencing of the expected amplicons confirmed that the obtained integration site and flanking sequence information were identical with those from PE resequencing analysis (Fig. 2c).

The results of WT resequencing data yielded 54,819,968 and 322 read pairs that mapped to the rice reference genome and the transformed plasmid, respectively (Table 1). A total of 262 read pairs were extracted for which one end mapped to the rice reference genome and the other mapped to the transformed plasmid using the original TranSeq pipeline. Using the modified TranSeq pipeline, 257 read pairs from the homologous sequence between the entire plasmid and rice reference genome were discarded and only five read pairs were retained as candidate reads (Table 1). Among these 257 read pairs, 17 pairs mapped to Chr 01, 90 mapped to Chr 02, nine pairs mapped to Chr 03, and 141 pairs mapped to Chr 10. The locations of these read pairs on rice genome were exactly the same as those 253 filtered out pairs in G281 (Table 2). Among the filtered five pairs, included four read pairs and one read pair that appeared to match Chr 02 and Chr 07, respectively (Supplementary Table 1). Further BLASTN analysis of these five pairs showed that all five were not really positives due to homologous sequences in transformation vectors or sequencing background noise. The WT resequencing results also confirmed its non-GMO authenticity, and the data were helpful for further identifying transgene integration in rice event G281.

**There are two copies of T-DNA inserted into rice genome.** To evaluate the copy number of the inserted transgenes based on the resequencing data, the formula $CopyNumber_{Seq.Data} = \frac{ADT_{gm}}{D_{gm} * R_{gm}} - \frac{ADT_{wt}}{D_{wt} * R_{wt}}$ (1) was used to calculate the copy number. In the formula, ADT represents the <u>A</u>verage sequencing <u>D</u>epth of the <u>T</u>arget gene, D represents the general sequencing depth, R is the relative relationship between mapped reads and the host genome, gm is the data of G281 line, and wt is the data of wide type Xiushui 110. To improve the accuracy of copy number evaluation and decrease the sequence bias in resequencing, the single copy rice endogenous reference gene, *SPS*, was used as a calibrator. The calculated copy number for the *G6 EPSPS* gene, the *hLF* gene, the rice *Gt1* promoter, the RNAi rice *CYP81A6* gene, the maize *Ubiquitin* promoter, and the maize PEPC terminator in G281 event was 2.14, 1.95, and 2.02, 2.00, 1.72, and 1.75, respectively (Table 3), indicating that there were two copies of each transferred into the rice genome.

Furthermore, the *G6 EPSPS* and *hLF* gene copy number were also analyzed using the ddPCR method with gene-specific primers and TaqMan probes to compare with the copy number from PE-WGS analysis. The copy number was calculated with the formula of $CopyNumber = \frac{transgene\ amount}{endogenous\ reference\ gene\ amount}$ (2). The ddPCR results showed that the copy number of *G6 EPSPS* and *hLF* genes was 1.87 and 1.96, respectively (Table 3). The results

**Table 2 Extracted read pairs crossing the transgene insertion site derived from original and modified TranSeq pipeline in G281 rice and WT.**

| | G281 | | Reason | WT (Xiushui 110) | | Reason |
|---|---|---|---|---|---|---|
| | Not filtered | Filtered | | Not Filtered | Filtered | |
| Chr 01 | 17 | 0 | Not really positives[a] | 17 | 0 | Not really positives[a] |
| Chr 02 | 75 | 2 | Not really positives[b] | 94 | 4 | Not really positives[b] |
| Chr 03 | 70 | 62 | candidate | 9 | 0 | Not really positives[b] |
| Chr 04 | 1 | 1 | noise sequences/alignment jitter | 0 | 0 | |
| Chr 06 | 1 | 1 | noise sequences/alignment jitter | 0 | 0 | |
| Chr 07 | 0 | 0 | | 1 | 1 | noise sequences/alignment jitter |
| Chr 08 | 1 | 1 | noise sequences/alignment jitter | 0 | 0 | |
| Chr 10 | 155 | 0 | Not really positives[a] | 141 | 0 | Not really positives[a] |
| Chr 11 | 1 | 1 | noise sequences/alignment jitter | 0 | 0 | |
| Chr 12 | 1 | 1 | noise sequences/alignment jitter | 0 | 0 | |

[a]Homologous sequences between *Gt1* promoter and rice genome.
[b]Homologous sequences between CYP81A6 RNAi and rice genome.

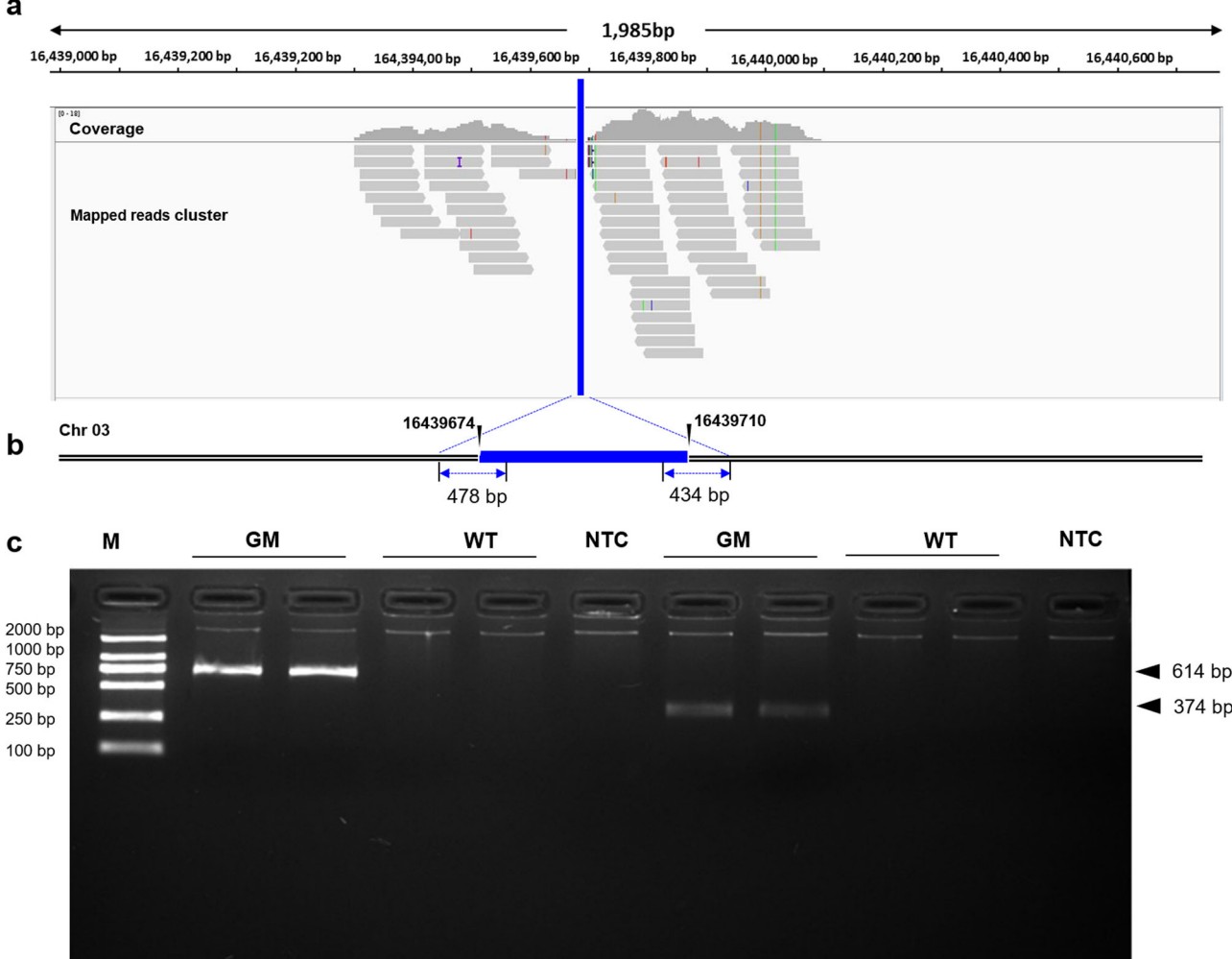

**Fig. 2 The transgene integration site and its flanking sequence from PE-WGS and PCR analyses. a** IGV view of 69 candidate read pairs crossing the insertion site of chromosome 3. The red line indicates the transgene insertion site in the rice genome. **b** Location of the transgene integration site according to the assembled contigs from candidate read pairs. **c** Conventional PCR amplification results for the flanking sequence. Lane M, DNA marker with six DNA fragments of 100, 250, 500, 750, 1000, and 2000 bp, respectively; Lane GM, G281; Lane WT, Xiushui 110; Lane NTC, no template control.

from both PE-WGS data and ddPCR suggested that the copy number of *G6 EPSPS* and *hLF* genes was two, and the copy number from PE-WGS analysis was slightly higher than that from ddPCR. Thus, based on the PE-WGS and ddPCR results, we concluded that two copies of plasmid T-DNAs were inserted into the rice genome.

**Arrangement and full sequencing of whole transgene integration**. A total of 2604 read pairs in which both ends were well mapped to the transformed plasmid were extracted for rice G281. These extracted reads were used to assemble transgene integration using SPAdes. A total of four contigs were obtained, including one with a length of 9145 bp, while the other three were shorter (279 bp, 266 bp, and 213 bp; Supplementary Sequence File 2). BLASTN results showed that only the 9145 bp contig corresponded to transgene integration, which contained three RNAi expression cassettes; one for *CYP81A6* including a 3′ untranslated region (UTR), the RNAi sequence, and the *CaMV35s* promoter; one for *G6 EPSPS* (*PEPC* terminator, *G6 EPSPS* gene, and *ubiquitin* promoter); and one for *hLF* (*PEPC* terminator, *hLF* gene, and rice *glutelin* promoter, *Gt1*) in a tandem arrangement (Fig. 3). Based on the copy number evaluation and the single transgene integration site in Chr 03, we concluded that the transgenic T-DNA inserted into rice Chr 03 locus 16,439,674 included two copies of complete T-DNAs. Furthermore, the flanking sequences adjacent to the integration site showed that both the 5′ and 3′ ends of rice genome DNAs were connected to the 3′ UTR of the RNAi expression cassette, indicating that the two T-DNAs were arranged head-to-head. The conventional PCR results also confirmed that these two copies of entire T-DNA sequences were inserted in a head-to-head rearrangement. A schematic diagram of the whole DNA insertion is shown in Fig. 3.

**No unintended integration and plasmid backbone residual insertion occurs in G281 event**. Unintended DNA integration and plasmid backbone residual insertions often occur during transgene transformations, due to the random nature of exogenous DNA integration, especially when using particle bombardment transformation. By mapping the cleaned raw NGS data for both GM and WT rice lines to the transformation plasmid sequence, we found nearly no integration or insertion of plasmid backbone residual sequences in G281, other than the T-DNA region in IGV view (Fig. 4). A comparison of the IGV view between WT and G281, no additional plasmid backbone residual sequences was observed in G281, although two sporadic sequences mapped to the plasmid Ori region were observed. The two sporadic sequences might come from the rice genome or sequencing contaminations considering the sequencing depth of ~29 X. We also performed conventional PCR analysis targeting the plasmid backbone sequences and did not obtain any positive amplicons from G281 DNA, further supporting the conclusion that there were no residual transformation plasmid backbone sequences in G281.

**The mimic Gt1 gene insertion and copy number were verified from the dataset of the artificial cisgenesis rice line**. As shown in Supplementary Fig. 3, the transferring T-DNA of rice G281 event contains three gene expression cassettes (the *hLF* expression cassette, the *G6 EPSPS* expression cassette, and the RNA interference cassette) in turn, and the DNA sequences of rice *Gt1* promoter and RNAi of *CYP81A6* gene connect with LB and RB, respectively. Considering the definition of cisgenesis/intragenesis, only the DNAs from native genome or a cross-compatible species should be transferred and inserted into recipient genome, the

**Table 3 Estimation of the copy number of transgenes in rice line G281 using resequencing data and ddPCR.**

| Gene | PE-WGS | | | | | | | ddPCR assay | | | | | |
| | G281 | | | WT | | | | Quantified copies | | | | | |
| | $ADT_{gm}$ | $D_{gm}$ | $R_{gm}$ | $ADT_{wt}$ | $D_{wt}$ | $R_{wt}$ | Copy number | 1 | 2 | 3 | Mean | RSD (%) | Copy number |
|---|---|---|---|---|---|---|---|---|---|---|---|---|---|
| HLF | 48.93 | 28.91 | 0.989 | 0 | 29.28 | 0.989 | 1.95 | 37160 | 37760 | 37150 | 37356.7 | 0.94 | 1.96 |
| EPSPS | 53.76 | 28.91 | 0.989 | 0 | 29.28 | 0.989 | 2.14 | 35680 | 35800 | 35320 | 35600 | 0.70 | 1.87 |
| Gt1 | 77.75 | 28.91 | 0.989 | 27.05 | 29.28 | 0.989 | 2.02 | / | / | / | / | / | / |
| CYP81A6 RNAi | 50.86 | 28.91 | 0.989 | 0.59 | 29.28 | 0.989 | 2.00 | / | / | / | / | / | / |
| Ubiquitin | 43.87 | 28.91 | 0.989 | 0.79 | 29.28 | 0.989 | 1.72 | / | / | / | / | / | / |
| PEPC | 43.97 | 28.91 | 0.989 | 0 | 29.28 | 0.989 | 1.75 | / | / | / | / | / | / |
| SPS | 25.11 | 28.91 | 0.989 | 25.14 | 29.28 | 0.989 | | 18980 | 19000 | 19100 | 19026.7 | 0.34 | |

*"Gt1" rice glutelin promoter. "CYP81A6 RNAi", including a 3' untranslated region (UTR), the RNAi sequence, and the CaMV35s promoter. "/" means did not detect.*

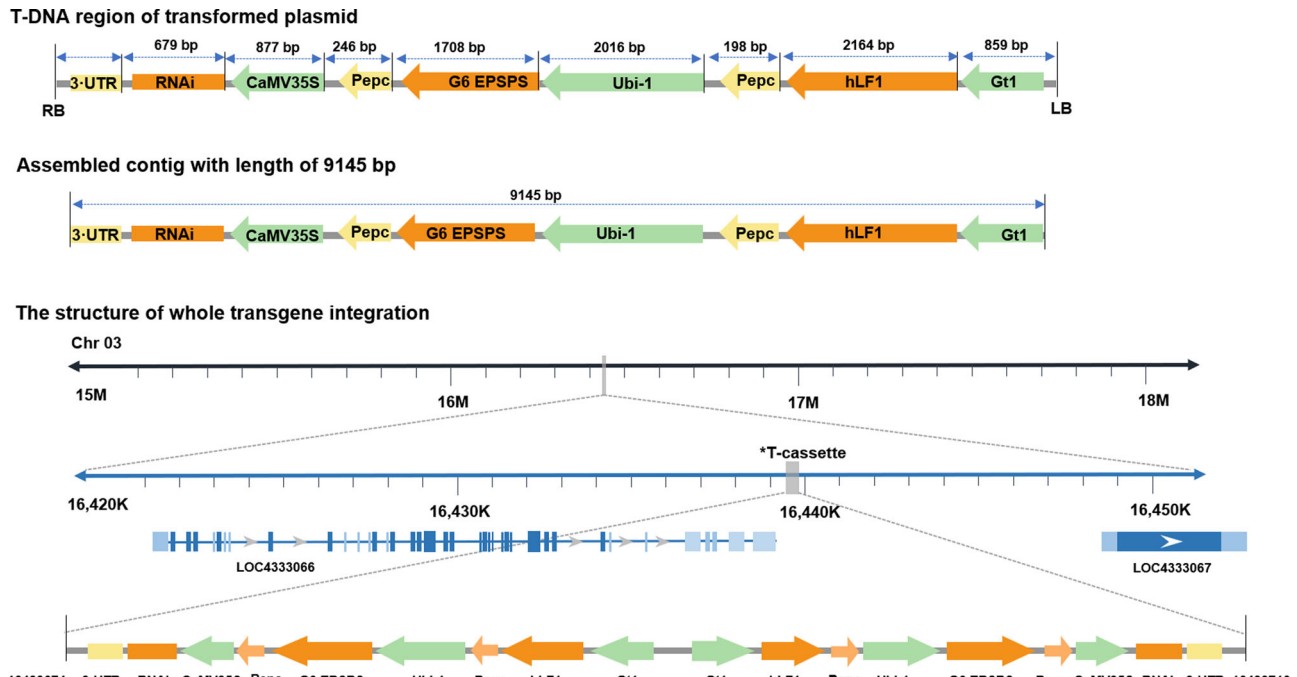

**Fig. 3 Schematic diagram of whole transgene integration in G281 rice.** RB, right border of T-DNA; LB, left border of T-DNA; *Gt1*, rice glutelin promoter; hLF, human lactoferrin; *G6 EPSPS*, the 5-enolpyruvylshikimate-3-phosphate synthase isolated from *Pseudomonas putida* fused with a corn chloroplast transit peptide at the N-terminus (gb: EU169459); RNAi, reverse repeat sequence for RNA interference against *CYP81A6*; CaMV35S, cauliflower mosaic virus 35S promoter; ZmUbi, polyubiquitin-1 promoter of *Zea mays*; PEPC, *phosphoenolpyruvate carboxylase* terminator of *Zea mays*.

DNAs normally were integrated at the other locus site. For identify the native DNAs inserted sites and flanking sequences of cisgenesis/intragenesis, the strategy is essentially to find the junctions between native DNAs and recipient genome DNA and to eliminate background noise from native DNAs of recipient genome, which is quite similar with that of G281 rice event analysis (Supplementary Fig. 3). Therefore, we speculated that the modified pipeline could be used for identifying the native DNAs inserted sites and flanking sequences of cisgenesis/intragenesis.

To prove our conjecture, we simulated an in silico mimicking cisgenesis event carrying one mimic *Gt1* gene insertion, since few real cisgenic crop lines were produced and commercialized and we can not obtain real cisgenesis materials. The artificial (simulated) dataset was created by in silico spiking of the non-GM line Xiushui 110 dataset with mimicked reads (Supplementary Data 1) corresponding to a 4152 bp contig containing 2100 bp rice genome sequence (Chr 5: 11030351-11031380, 11031381-11032420) and 2052 bp of rice *Gt1* gene. (Supplementary Fig. 4 and Supplementary Sequence File 3). The in silico mimicking cisgenesis event only contained the mimic rice *Gt1* insert without any elements from other species. For identifying the mimic rice *Gt1* insert, the *Gt1* sequence with 2052 bp was used as plasmid reference sequence. With the pipelines, a total of 1382 read pairs were extracted as candidates using the original TranSeq pipeline. However, only 271 read pairs was obtained as candidates using our modified TranSeq pipeline, and 1111 read pairs from homologous sequence of rice genome were discarded (Table 4). Among the 1111 discarded read pairs, 103 pairs mapped to Chr 01 and 1008 pairs mapped to Chr 10. All the sequences of these read pairs were from the homologous sequences between *Gt1* gene and rice genome. We also found this kind of reads in WT line as well. Among the 271 potential candidate reads, 33 read pairs mapped to Chr 03, one pair mapped to Chr 04, 116 pairs mapped to Chr 05, and 121 pairs mapped to Chr 10. After considering the sequencing

depth (29.28×) and BLASTN analysis, the 116 read pairs mapped to Chr 05 were confirmed as true candidates (Table 4). The other 155 read pairs mapped to other chromosomes were also confirmed to be untrue, because the BLASTN analysis showed that those reads were still from the homologous sequences between *Gt1* gene and rice genome (Table 4). IGV visualization of the 116 candidate read pairs showed that only one transgene integration was introduced at locus 11,031,380 of Chr 05 (Supplementary Fig. 4). The assembled contigs (Supplementary Sequence File 4) generated from the 116 read pairs in de novo analysis clearly showed the T-DNA integration site at Chr 05 locus 11,031,380, and two contigs with lengths of 408 bp and 406 bp comprising the 5′ and 3′ end flanking sequences adjacent to transgene insertion. In WT resequencing data analysis, a total of 766 and 155 read pairs were extracted using the original TranSeq pipeline and the modified TranSeq pipeline, respectively. A total of 622 read pairs were discarded and 155 read pairs were still retained as candidate reads (Table 4). The discarded reads and candidate reads in WT line could be observed from those in the artificial cisgenesis rice line, indicating that the 155 candidate read pairs were the background noise from the WT line, and these reads were generated from the he homologous sequences between *Gt1* gene and rice genome.

According to the formula of copy number evaluation, the copy number of *Gt1* insertion was also calculated. The calculated copy number for the *Gt1* gene (2052 bp) was 1.17 (Supplementary Table 2), indicating that there was one copy of *Gt1* gene transferred into the rice genome.

## Discussion

Clear molecular characterization of GM events is required to approve GMOs and GM products for commercialization, including new transgenic crops produced using DNA recombinant techniques

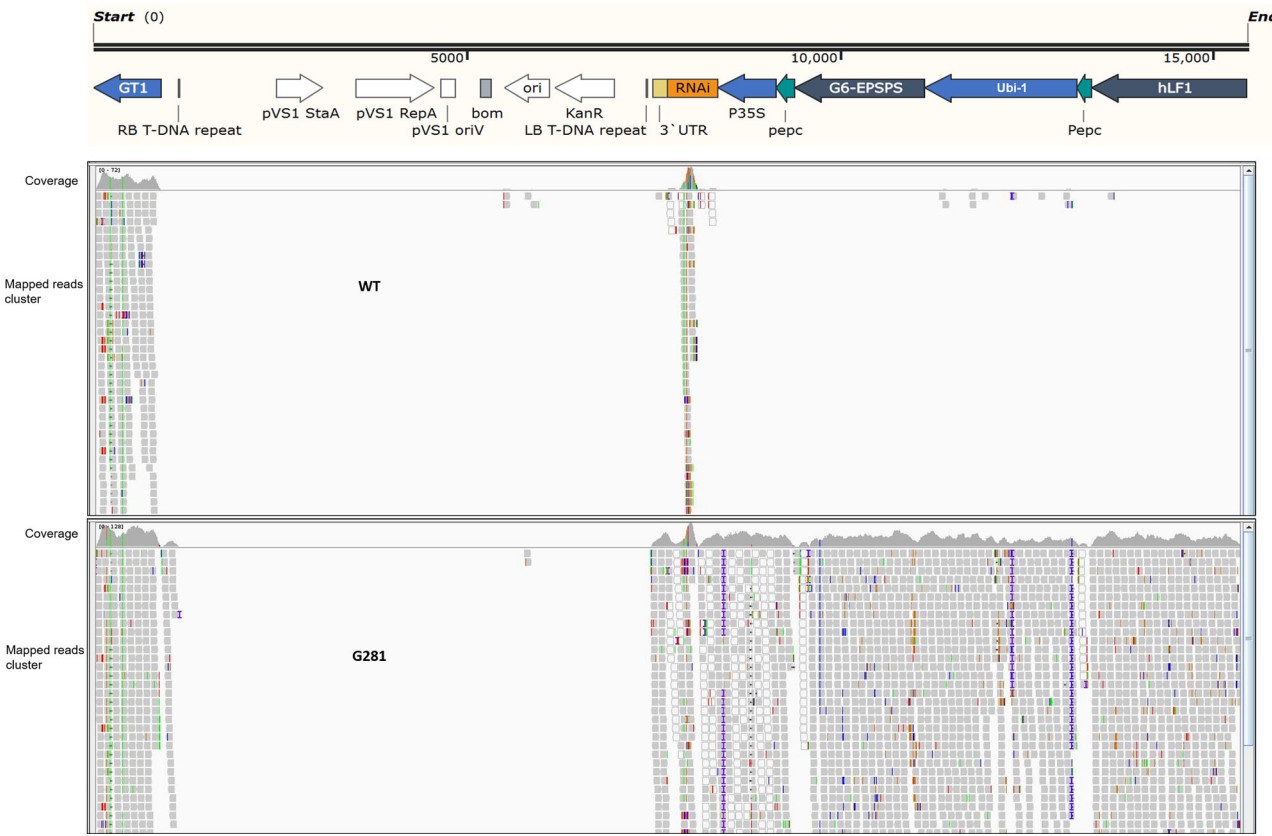

**Fig. 4 IGV of the reads mapped to transformed plasmid.** Read pairs mapped to the entire transformed plasmid from WT and G281 event paired-end sequencing data.

**Table 4 Extracted read pairs crossing the *Gt1* inserted site derived from original and modified TranSeq pipeline in artificial cisgenesis rice and WT.**

|  | Artificial cisgenesis | | Reason | WT (Xiushui 110) | | Reason |
|---|---|---|---|---|---|---|
|  | Not filtered | Filtered |  | Not filtered | Filtered |  |
| Chr 01: 32076358-32077341 | 20 | 20 | Not really positives[a] | 20 | 20 | Not really positives[a] |
| Chr 01: 32076840-32078305 | 103 | 0 | Not really positives[a] | 103 | 0 | Not really positives[a] |
| Chr 01: 32078413-32078798 | 13 | 13 | Not really positives[a] | 13 | 13 | Not really positives[a] |
| Chr 04: 675093 | 1 | 1 | noise sequences/alignment jitter | 1 | 1 | noise sequences/alignment jitter |
| Chr 05: 11030976-11031684 | 116 | 116 | candidate | 0 | 0 |  |
| Chr 10: 13497001-13497784 | 65 | 65 | Not really positives[a] | 65 | 65 | Not really positives[a] |
| Chr 10: 13497378-13499421 | 1008 | 0 | Not really positives[a] | 508 | 0 | Not really positives[a] |
| Chr 10: 13499431-13499725 | 56 | 56 | Not really positives[a] | 56 | 56 | Not really positives[a] |

[a] Homologous sequence between *Gt1* and rice genome.

such as cisgenesis and intrasgenesis[21]. For a comprehensive understanding of transgene integration, NGS has been often accepted and employed to decipher the transgene integration of GMOs. Advantages of NGS include higher efficiency, accuracy, and reliability than with traditional sequencing methods[22]. However, most published research has focused on transgene integration in conventional transgenic crops/animals, and few studies have explored the possibility of using NGS approaches for molecular characterization of cisgenesis and intragenesis since distinguishing and identifying the cis gene from native recipient genome background is quite difficult. In the present work, the G281 transgenic rice event was generated by transforming a plasmid containing three gene expression cassettes. In the plasmid, the 5′ end of T-DNA is connected to the RNAi sequence of the rice *CYP81A6* gene, and its 3′ end is connected to the rice *Gt1* promoter.

According to the definition of cisgenesis/intrasgenesis and the principle of identifying transgene insertion with PE-WGS approach, the G281 line might be regarded as having a structure similar to that acquired by cisgenesis/intragenesis, although the exogenous *G6 EPSPS* and hLF genes in T-DNA are derived from other species[23]. In the present work, we tried to develop a suitable pipeline to characterize transgene integration of G281 rice, and to evaluate the potential of characterizing transgene integration in cisgenesis/intrasgenesis using an in silico mimicking cisgenic line as an example. We performed successful molecular characterization of G281, including the transgene insertion site and its flanking sequences, copy number of transgenes, unintended insertions, and transformed plasmid backbone residues. Also, we successfully identified the transgene insertion and it copy number in the artificial cisgenic line.

The DNAs of both ends of the T-DNA region are from the rice host genome in G281, making it difficult to identify the integration site. Using previously developed pipelines, numerous read pairs from the rice host genome are extracted, decreasing the efficiency and accuracy of screening and the identification of candidate reads. Herein, 322 read pairs were extracted as candidates from G281 data using the TranSeq pipeline. However, only 69 read pairs remained when we used the modified pipeline with an additional step restricting the positions ± insert sizes of homologous sequences. According to the filtering of read pairs, the transgene integration site was easily identified with high accuracy, whereas more than 75% of extracted reads were not really positives using the original TranSeq pipeline[5]. These results indicate that the modified pipeline avoided the influence of host genome DNA effectively.

To determine the sequence and structure of transgene integration, we combined the results of copy number evaluation and de novo analysis, revealing one entire transgene integration at rice Chr 03 involving two repeated T-DNA regions of transformed plasmid, and the two repeats are arranged in a head-to-head manner. In general, the whole transgene integration could be assembled based on reads mapped to the transformed plasmid. However, this is difficult when multiple copies of T-DNAs are inserted. Since the reads from NGS data are very short, identifying DNAs with multiple repeats is challenging[11]. In G281, the whole transgene integration was not obtained directly from de novo analysis. We only assembled one repeat of the T-DNA region with a length of 9145 bp. Based on the sequences of flanking sequences and data from PCR amplification of the links between two repeats, we confirmed the whole structure and sequence of the transgene integration. The result of copy number evaluation corrected the previous reported result of only one single copy of transgene insertion in G281 line from Southern blotting analysis[19]. Some recent reports have used third-generation sequencing (TGS) approaches to reveal whole transgene integration[23–27]. However, analyzing whole transgene integration involving complex structures and/or rearrangements remains difficult using TGS approaches because of the absence of effective data analysis pipelines and quite high costs.

Molecular characterization of the G281 line using the developed pipeline showed that the approach is potentially suitable for investigating cisgenesis/intragenesis lines, in particular for identifying the transgene inserted site, flanking sequence, and copy number. We successfully proved the potential by employing an in silico mimicking cisgenesis rice carrying rice Gt1 gene insertion. In the artificial cisgenesis rice, we obtained 271 candidate read pairs around the transgene inserted site, and identified that there was only one transgene inserted into the rice genome at the locus 11,031,380 of Chr 05. The flanking sequences of inserted site were also assembled, and the copy number of Gt1 gene insertion was calculated with the value of 1.17 as well. Although our results confirmed that the modified pipeline is suitable for charactering the cisgenesis/intragenesis, control lines are more important and should be sequenced at the same time when exploring cisgenesis/intragenesis lines. By comparing NGS data between artificial cisgenesis and control lines, most reads (~80%) from the homologous native sequence can be filtered out using dedicated pipelines. For example, there were 155 read pairs extracted for the WT line and identified as potential candidates for determining the transgene integration site. Obviously, there were no transgenes transferred in the WT line, hence these 155 read pairs must be derived from homogenous sequences between the host genome and the transformed gene, sequencing bias, or sequencing data contamination. According to the reads extracted from the WT line, we also found reads that were similar to those in artificial cisgenesis, hence the similar reads were not really positives.

Therefore, control lines are essential for analysis of cisgenesis/ intragenesis lines. In the artificial analysis, it is difficult to assemble the whole inserted Gt1 gene or Gt1 insertion because of the high background noise from recipient rice genome. For example, a total of 155 read pairs were from the rice background noise in the final filtered 271 candidate read pairs, which will interfere with the further sequence assembly. Therefore, deciphering the full sequence and structure of whole transgene insertion in cisgenesis/intragenesis lines still need to be concerned in further research.

In summary, we modified our previously established TranSeq bioinformatics pipeline, and performed full molecular characterization of GM rice G281 using a PE-WGS approach with the modified pipeline. We identified only a single integration site in Chr 03 with two head-to-head repeats of complete T-DNAs inserted in G281. No residual transformation plasmid backbone sequences or unintended integration of exogenous DNA was detected in G281. Furthermore, we proved the potential of our modified pipeline in molecular characterization analysis of an artificial cisgenic rice line. The results confirm that the PE-WGS and modified TranSeq pipeline approach can be successfully applied to characterize transgene integration in transgenesis, cisgenesis, and intragenesis lines.

## Methods

**Plant materials**. Transgenic rice event G281 is associated with high sensitivity to bentazon, high tolerance to glyphosate, and high expression of human lactoferrin (hLF) protein in seed, according to Zhejiang University, China. T-DNAs of the binary vector p1300-S450RNAi-hLF-G6, including tandem arrays of three expression cassettes of *hLF*, *G6 5-enolpyruvylshikimate-3-phosphate synthase* (*G6 EPSPS*), and the RNAi of rice *CYP81A6* gene were introduced into the non-GM line Xiushui 110 via Agrobacterium-mediated transformation[23]. Seeds of transgenic rice event G281 and its recipient line (Xiushui 110) were kindly supplied by the developer.

**DNA extraction**. GM rice G281 event and its recipient line were grown in a greenhouse in Shanghai Jiao Tong University, China, and fresh leaves were sampled and ground into powder in liquid nitrogen for DNA extraction. Rice genomic DNA was extracted and purified using a Plant Genomic DNA Kit (Cat. no. DP305-3; TIANGEN Company, China). The quality and quantity of extracted DNA was evaluated with a Nanodrop ND-8000 spectrophotometer (Thermo Fisher Scientific, Waltham, MA, USA) and by 1% agarose gel electrophoresis, respectively. Purified DNA samples were stored and used for PCR, droplet digital PCR (ddPCR), and PE library construction.

**Paired-end whole genome sequencing (PE-WGS)**. Approximately 5 μg of extracted genomic DNA from GM G281 and wild-type line Xiushui 110 was used for PE library construction with an Illumina Paired-end Preparation Kit (Illumina Inc, USA) with an average insert size of 500 bp in length. Library quality was evaluated by Pico-Green (Quant-iT; Invitrogen, USA) and an Agilent Bioanalyzer DNA 1000 kit (Agilent Technologies, USA). Constructed libraries were subjected to sequencing on an Illumina HiSeq 2000 DNA sequencer (Illumina Inc, USA) by BGI-Shenzhen (Shenzhen, China), and 100 bp PE reads were generated. Raw sequencing data were preprocessed with Trimmomatic to remove index sequences, short and low-quality reads, and read lengths were unified. The resulting 100 bp trimmed sequencing read data were used for further analysis.

**In silico mimicking of one cisgenesis rice line carrying native Gt1 gene**. To test the potential for identifying of native gene insertion in cisgenesis/intragenesis line, a hypothetical cisgenesis line was designed and mimicked by inserting partial Gt1 gene with length of 2052 bp into rice genome at the site of Chr 05 11031380. The mimic integration of Gt1 gene has 4152 bp, containing three fragments of 1030 bp (rice Chr 05: 11030351-11031380), 2052 bp (Partial rice Gt1 gene), and 1040 bp (rice Chr 05: 11031381-11032420), and the detailed structure is described in Supplementary Fig. 5 and Supplementary Sequence File 3. Based on the sequence of mimic integration of Gt1 gene, a set of paired-end reads was generated randomly generated in silico using art-illumina software, corresponding to the same sequencing depth (29.28×) with the clean data of non-GM rice Xiushui 110 (As shown in Supplementary Data 1). The base error rate of each read was set to meet the uniformly decreasing distribution for an error rate of 0.1% at the start of the read increasing to 4% at the end of the read, and 5% probability to have a random DNA read, maximum 1 N allowed in a given read, the insertion size is 500 bp (mean distance between ends of read pairs) with the standard deviation of 10 bp. The reference mimic was defined as to have a mutation rate of 0.001, the fraction of

mutations that are indels was defined to 0.1, indel extension probability was set to 0.3, and the minimum length indel was set to 1 bp. The raw data of hypothetical cisgenesis line was prepared by mixing the generated paired-end reads with the clean sequencing data of non-GM rice (Xiushui 110) for further analysis.

**Bioinformatics pipeline for PE-WGS data analysis**. PE-WGS data were analyzed using bioinformatics pipeline of TranSeq with slight modifications (As illustrated in Fig. 1) to add functionality and avoid reads from the homologous sequence of host native genome DNAs[5]. To reduce interference from native genome DNAs, the reads from native sequence were filtered by three steps. First, BLASTN was used to search homologous sequences between the entire plasmid sequence and the recipient reference genome sequence[28]. Second, the position of similar sequences was added or subtracted 500 (for paired-end sequencing) to eliminate reads from native sequence. Finally, subtracting the blocking position of the TranSeq results to generate candidate results. Burrows-Wheeler Aligner (BWA version 0.6.2)[29], SPAdes Genome Assembler (SPAdes version 3.11)[30], Sequence Alignment/Map tools (SAM tools version 1.6)[31], and Integrative Genomics Viewer (IGV, version 2.4)[32] formed the pipeline for read mapping, assembly, alignment storage, and visualizing mapping results, respectively. Transformation plasmid sequences of G281 and rice reference gene sequences (RefSeq assembly accession: GCF_001433935.1) were used as reference sequences in the analysis.

**Conventional PCR and Sanger sequencing**. Primer pairs used for verifying transgene insertion sites and their flanking sequences were designed based on the candidate read pairs identified in PE-WGS analysis (Supplementary Table 3). Conventional PCR was performed with a Veriti thermocycler (ThermoFisher Scientific) in a total volume of 50 μl, including 5 μl of 10× Ex Taq Buffer (Takara, Japan), 5 μl of dNTPs (2 mM), 0.25 μl of Ex Taq polymerase (Takara), 2 μl of 10 μM forward primer, 2 μl of 10 μM reverse primer, 2 μl of DNA template (10 ng/μl), and 33.75 μl of RNase-free and DNase-free water. Each reaction was performed in triplicate, and thermal cycling included 5 min at 98 °C, followed by 35 cycles of 30 s denaturation at 94 °C, 30 s annealing at 55 °C, and 1 min extension at 72 °C, then an additional extension step at 72 °C for 7 min. Amplified products were purified, sent to Invitrogen (Shanghai, China) for Sanger sequencing, and the sequencing results were used as query sequences for BLASTN searching[33].

**Droplet digital PCR**. The ddPCR primers and probes for *G6 EPSPS* and *hLF* genes were designed using beacon designer software version 8.0, and our previously reported primers and probes for rice endogenous reference gene *SPS* were also used[34]. All primers and probes were purchased from Invitrogen (Shanghai, China) and are listed in Supplementary Table 4. The ddPCR assays were performed on a QX200 Droplet Digital PCR platform (Bio-Rad, Pleasanton, CA, USA) with a final volume of 20 μl comprising 10 μl of 2× ddPCR Supermix (Bio-Rad, Pleasanton, CA, USA), 1 μl of 10 μM forward primer, 1 μl of 10 μM reverse primer, 0.5 μl of 10 μM probe, 1 μl of DNA template with the concentration of 10 ng/μl, and 6.5 μl of RNase-free and DNase-free water. After preparation of the 20 μl reaction mix for ddPCR, droplets were generated in 8-well cartridges using a QX200 droplet generator (Bio-Rad, Pleasanton, CA, USA). Water-in-oil emulsions were transferred to a 96-well plate and amplified in a T100 PCR cycler (Bio-Rad, Pleasanton, CA, USA). The procedure for ddPCR assays included 5 min at 95 °C, followed by 40 cycles of a two-step thermal profile comprising 30 s at 95 °C and 60 s at 58 °C at a ramp rate of 2.0 °C/s. After cycling, each sample was incubated at 98 °C for 10 min then cooled to 4 °C. After PCR amplification, PCR plates were transferred to a QX200 droplet reader (Bio-Rad, Pleasanton, CA, USA), and data acquisition and analysis were performed using QuantaSoft (Bio-Rad, Pleasanton, CA, USA).

**Statistics and reproducibility**. In conventional PCR, each sample were performed with two parallels. In ddPCR, all reactions were repeated in triplicate with three parallels. The reproducibility of ddPCR experiments was evaluated with relative standard variation.

**Reporting summary**. Further information on research design is available in the Nature Research Reporting Summary linked to this article.

## Data availability

The raw re-sequencing data of GM rice G281 event and non-GM rice line Xiushui 110 are available in Sequence Read Archive of National Center for Biotechnology Information (Accession Number: SRR18236702 and SRX3923908).

## Code availability

BWA version 0.6.2 (https://sourceforge.net/projects/bio-bwa/files/bwa-0.6.2.tar.bz2/download); SPAdes version 3.11 (https://github.com/ablab/spades/releases/tag/v3.11.0); SAM tools version 1.6 (https://sourceforge.net/projects/samtools/files/samtools/1.6/); IGV, version 2.4 (https://software.broadinstitute.org/software/igv/ReleaseNotes/2.4.x).

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

## Acknowledgements

The authors greatly thank Congmao Wang, Sheng Quan, and Jianxin Shi for kindly comments on the manuscript preparation. This work was supported by the National Transgenic Plant Special Fund (Grant No. 2016ZX08012-002), and the Program for Professor of Special Appointment (Eastern Scholar) at Shanghai Institutions of Higher Learning.

## Author contributions

W.X., H.Z., and Y.Z. performed the experiments, analyzed the data, and draft the manuscript; R.L. P.S., and X.L. helped to revise and comment on this work. L.Y. conceived and designed the experiments, write the manuscript. All authors have read and agreed to the published version of the manuscript.

## Competing interests

The authors declare no competing interests.
