## [Peer Review File · Communications Biology]

Reviewers' comments:

Reviewer #1 (Remarks to the Author):

Strengths of the paper:

- The authors laid out a clear and precise approach to identifying the exact location of a transgene using paired-end whole genome sequencing.
- Coupling PE-WGS with PCR and digital droplet ensured precision in identification and location of the transgene in the rice genome chromosome 3.

Major concern: The basis of the paper's strength is not factual

- Authors have based the strength of their approach in its effectiveness in identifying intragene/cisgene in GM rice event G281. They assert that this line is "similar to" cisgenic/intragenic. However, this rice line is not cisgenic/intragenic and the argument of it being "similar to" cisgenic/intragenic is not a logical one. This assertion is obviously misleading. Authors should have used an actual cisgenic/intragenic line if they wanted to front this argument as the paper's strength. Authors should therefore clearly indicate that GM rice event G281 is transgenic and change this in the running title, abstract and within the text and have the paper evaluated on that basis. The GM rice event G281 has genes from Maize, human and *Pseudomonas putida* making it transgenic.
- In their own words, authors agree that investigating intragenesis is more complex than line G281 (which is essentially transgenic) – lines 343-345.

Other concerns

- In the materials and methods (lines 143 – 146), authors indicate that they filtered false-positive reads by searching for positions of similar sequences between plasmids and references using BLASTN. However, they did not indicate the results of this comparison in the results section. This is important since some plasmid sequences which authors considered false positives were observed in the transgenic and non-transgenic wildtype (lines 315-320 and 348 - 350).
- Authors indicate that the sequences homologous to transgenic ones in the wildtype "...must be derived from homogenous sequences between the host genome and the transformed plasmid, sequencing bias, or sequencing data contamination" – Lines 348 – 355. If this approach is to be solid enough, authors need to be sure where these sequences came from rather than guess.

Reviewer #2 (Remarks to the Author):

Overall, the manuscript is in good shape. The language is adequate and easy to read. Although its novelty is not very strong, the authors' demonstrated that PE-WGS is a useful tool for molecular characterization analysis of cisgenesis/intragenesis crops. This is valuable to the relevant scientific community. In order to make the conclusion more reasonable, the authors should make some revision and explanation.

- 1) for 3.2, how 69 read pairs that were retained as candidate reads for determining the T-DNA insertion site distributed? How much on 3' junction region, how much on 5' junction region. How to determine that the sporadic read pairs mapped to other chromosomes might be false positive reads? It should be proved by experiment.
- 2) for 3.3 copy number of exogenous DNA. There only analysis the copy number of foreign gene G6-EPSPS and hlf. Please add how to analysis copy of RNAi of rice CYP81A6 gene, because the purpose of this article is that the using of PE-WGS on molecular characterization analysis of

cisgenesis/intragenesis crops.

3) There should have the IGV analysis of between 69 read pairs that were retained as candidate reads for determining the T-DNA insertion site and T-DNA sequence.

4) How to know there have not backbone insertion from Figure 4.

5) DNA extraction in 2.1 should be put at 2.2 part.

6) please revised some mistakes such as:

(1) Line116, xiushui-110, Line117 xiushui 110, please full text unification

(2) Line 115, gene should be orthographic

(3) Line 126, paired-end whole-genome sequencing should be deleted.

(4) Line 170 hFL is hlf gene

(5) in Table 2,there should have CYP81A6 analysis.

Point to point responses to the Reviewers' comments

Responses to comments from Reviewer #1

Reviewer #1 (Remarks to the Author):

Strengths of the paper:

- The authors laid out a clear and precise approach to identifying the exact location of a transgene using paired-end whole genome sequencing.
- Coupling PE-WGS with PCR and digital droplet ensured precision in identification and location of the transgene in the rice genome chromosome 3.

Comment 1: Major concern: The basis of the paper's strength is not factual

- Authors have based the strength of their approach in its effectiveness in identifying intragene/cisgene in GM rice event G281. They assert that this line is "similar to" cisgenic/intragenic. However, this rice line is not cisgenic/intragenic and the argument of it being "similar to" cisgenic/intragenic is not a logical one. This assertion is obviously misleading. Authors should have used an actual cisgenic/intragenic line if they wanted to front this argument as the paper's strength. Authors should therefore clearly indicate that GM rice event G281 is transgenic and change this in the running title, abstract and within the text and have the paper evaluated on that basis. The GM rice event G281 has genes from Maize, human and *Pseudomonas putida* making it transgenic.

Answer: Thank you very much for pointing out the inaccuracies in our description on the G281 rice event. In the section of "Plant material", we introduced the detail information of transgenic

rice event G281, which was produced by inserting the T-DNA containing three expression cassettes of *hLF*, *G6 EPSPS*, and the RNAi of rice *CYP81A6* gene into the non-GM line 115 Xiushui-110 via *Agrobacterium*-mediated transformation. Since the elements and exogenous genes from other species existed into the T-DNA region, the G281 rice is transgenic. According to the definition of *cisgenic/intragenic* line, all the elements should be from native genome, therefore the G281 rice is not a real *cisgenic/intragenic* line. We have clearly indicated that GM rice event G281 is transgenic and changed the running title in the whole MS.

In our work, we aim to explore the molecular characterization of transgenic rice G281 event, which was produced by introducing the T-DNAs. The T-DNA contains three gene expression cassettes (the *hFL* expression cassette, the *EPSPS* expression cassette, and the RNA interference cassette) in turn. Initially, we failed to identify the inserted sites and flanking sequences using our previously developed pipeline of TranSeq. Then, we found that both end of DNA sequences connected with LB and RB in T-DNA were from the rice recipient genome (rice *Gt1* promoter and RNAi of *CYP81A6* gene). It is the reason for failure with TranSeq analysis. Therefore, we modified and update the pipelines of TranSeq to remove the background noise from the native gene or DNA sequences of rice *Gt1* promoter and *CYP81A6* gene, and successfully found the T-DNA inserted sites and flanking sequences. Considering the definition of *Cisgenesis/intragenesis*, only the DNAs from native genome or a cross-compatible species should be transferred and inserted into recipient genome, the DNAs normally were integrated at the other locus site. For identify the native DNAs inserted site and flanking sequences of *Cisgenesis/intragenesis*, the strategy is essentially to find the junctions between native DNAs and recipient genome DNA and to eliminate background noise from native DNAs of recipient genome, which is quite similar

with that of G281 rice event analysis (As shown in the following figure). Therefore, we believe that the modified pipeline could be used for identifying the native DNAs inserted sites and flanking sequences of *Cisgenesis/intragenesis*.

In *cisgenic/intragenic* line, the DNA sequences of 5' and 3' end of T-DNA are both from native genome. In G281 line, the DNA sequences of 5' and 3' end of T-DNA (rice Gt1 promoter and RNAi of CYP81A6 gene) are also from rice genome. This is the reason why we used the words of "similar to". Now we know that the words of "similar to" is not rigorous with your kindly comment. In the new revision, we have removed any mentioning of *Cisgenesis/intragenesis* in the title and the main text that may mislead readers to think G281 is a *cisgenesis/intragenesis* line.

According to your comment, we have contacted many collaborators working on breeding new crop lines with new breeding techniques, but we failed to find a true *cisgenesis/intragenesis* line. Although we did not have one real *Cisgenesis/intragenesis* line, we hope to verify it using simulated NGS data generated randomly from one mimic rice line with native *Gt1* gene insertion in the revised version. The result showed that the mimic *Gt1* gene insertion could be identified with the detail inserted site, flanking sequences, and copy number (**Table 4; Supplemental File 1, 4; Supplemental Figure S1 S3, and S4; supplemental Table S1 and S5**). We

believe that these results can do favor to confirm that our developed pipelines were helpful for the analysis of *cisgenesis/intragenesis* in theory.

Nevertheless, one point we are trying to make in the original and the revised MS is that we believe that some of the analyses we did in the G281 study could potentially be useful in analyzing a true *cisgenesis/intragenesis* line, and I do agree with you that we didn't clearly state that G281 is not a *cisgenesis/intragenesis* line, and we hope you will find that in the revised MS we paid great attention to this concern and hope that the revised text could make our point more accurate.

Comment 2: - In their own words, authors agree that investigating intragenesis is more complex than line G281 (which is essentially transgenic) – lines 343-345.

Answer: In general, full molecular characterization of transgenic crops contains the information of exogenous DNA insertion site, flanking sequence, whole exogenous DNA arrangement and sequence of the insertion, and copy number of inserted DNAs. However, our developed pipelines have the potential in identifying native DNAs inserted site and flanking sequences of *cisgenesis/intragenesis*. It is still challenging in obtaining the whole exogenous DNA arrangement and sequence of the insertion in *cisgenesis/intragenesis* using our modified pipelines. That is why we said that investigating full molecular characterization of *cisgenesis/intragenesis* is more complex than line G281 in the manuscript.

Other concerns

Comment 3:- In the materials and methods (lines 143 – 146), authors indicate that they filtered false-positive reads by searching for positions of similar sequences between plasmids and references using BLASTN. However, they did not indicate the results of this comparison in the results section. This is important since some plasmid sequences which authors considered false positives were observed in the transgenic and non-transgenic wild type (lines 315-320 and 348 - 350).

Answer: Thanks for your kind comment. As shown in Table 1, a total 257 reads were filtered from 262 candidate reads in non-transgenic wild type, and a total 253 reads were filtered from 322 candidate reads in transgenic G281 reads by searching for positions of similar sequences between plasmids and references using BLASTN in modified pipeline, these results indicated that most of the false positive reads could be effectively culled out, which will do great favor to identify the real inserted site and flanking sequencing and to decrease further experimental confirmation. Also, we have added the details of comparison in **Table2** in the revised version.

Comment 4:- Authors indicate that the sequences homologous to transgenic ones in the wild type “...must be derived from homogenous sequences between the host genome and the transformed plasmid, sequencing bias, or sequencing data contamination” – Lines 348 – 355. If this approach is to be solid enough, authors need to be sure where these sequences came from rather than guess.

Answer: Thanks for your kind comment. In the new revision, we have listed all the 262 read pairs and described the details of where they came from in **Table 2**.

Reviewer #2 (Remarks to the Author):

Overall, the manuscript is in good shape. The language is adequate and easy to read. Although its novelty is not very strong, the authors' demonstrated that PE-WGS is a useful tool for molecular characterization analysis of cisgenesis/intragenesis crops. This is valuable to the relevant scientific community. In order to make the conclusion more reasonable, the authors should make some revision and explanation.

Comment 1: for 3.2, how 69 read pairs that were retained as candidate reads for determining the T-DNA insertion site distributed? How much on 3' junction region, how much on 5' junction region. How to determine that the sporadic read pairs mapped to other chromosomes might be false positive reads? It should be proved by experiment.

Answer: Thanks for your kind comment. We have added the details of 69 read pairs in Table 2 in the revised version. Among the 69 read pairs, a total of 22 read pairs covered the 5' junction region, a total of 40 read pairs covered the 3' junction region, and the other seven read pairs covered different loci which were considered and verified as false positive reads. In our work, the mean sequencing depth was 28.91, which indicated that each nucleotide should be sequenced with ~28.91 times in theory. For example, a total of 22 read pairs were observed around the 5' junction region, and a total of 40 read pairs were observed around the 3' junction region. However, the other seven read pairs showed seven different loci, which meant that only one read pair was observed in each locus, therefore, we speculated that those seven read pairs were false positive. Also, we have designed the primers according to the seven read pairs and verified them using PCR, the results showed that two read pairs have no amplification in both

G281 line and WT line, five read pairs have the same amplification between G281 line and WT line, indicating that all the seven read pairs were false positive. All the results of PCR analysis were shown in **Supplemental Figure S2** in the new revision.

Comment 2: for 3.3 copy number of exogenous DNA. There only analysis the copy number of foreign gene G6-EPSPS and hlf. Please add how to analysis copy of RNAi of rice CYP81A6 gene, because the purpose of this article is that the using of PE-WGS on molecular characterization analysis of cisgenesis/intragenesis crops.

Answer: Thanks for your kind comment. We have evaluated the copy number of the exogenous genes (G6-EPSPS and HLF), native gene (RNAi of rice CYP81A6 gene), native elements (Gt1 promoter), and exogenous elements (maize ubiquitin promoter and maize PEPC promoter). The calculated copy number were shown in Table 3. For calculating the copy number of native gene or element, the copy number from the formula $CopyNumberSeq.Data = ADT / (D * R)$ contains the value of native genome besides the inserted gene or element. Therefore, the copy number of native gene or element should be further calculated by the formula of $Cp = Cp_{GM} - Cp_{wt}$. We have revised the copy number analysis of native gene/element in the new revision.

Comment 3: There should have the IGV analysis of between 69 read pairs that were retained as candidate reads for determining the T-DNA insertion site and T-DNA sequence.

Answer: Thanks for your kind comment. We have showed the IGV view of the 62 read pairs covering the inserted region in **Figure 2A** in original MS. The IGV view of other seven false positive read pairs were also performed, however, IGV view showed that all seven pairs can not be visualized within reasonable distance or locations according to rice reference genome (RefSeq assembly accession: GCF_001433935.1. The sequence information of the other seven false positive read pairs were also listed in **supplemental Table S4**.

Comment 4: How to know there have not backbone insertion from Figure 4.

Answer: Thanks for your kind comment. The above image in **Figure 4** showed the plasmid diagram including backbone and T-DNA, and the whole sequences of plasmid were used to IGV analysis. The middle image in **Figure 4** was IGV view of the reads from WT line, which showed almost all the sequenced reads of WT line only matched to the rice *Gt1* promoter and RNAi of rice *CYP81A6* gene, and few reads matched to the backbone region of plasmid. The low image in **Figure 4** was IGV view of the reads from G281 line, which showed that almost all the sequenced reads of G281 line only matched to T-DNA region of plasmid, and few reads matched to the backbone region of plasmid. Combined the IGV view results of G281 and WT, we believed that no backbone sequence of plasmid were inserted into the G281 line.

Comment 5: DNA extraction in 2.1 should be put at 2.2 part.

Answer: Thanks for your kind comment. We have revised it in the new MS.

Comment 6: please revised some mistakes such as:

(1) Line116, xiushui-110, Line117 xiushui 110, please full text unification

Answer: Thanks for your kind comment. We have revised it in the new MS.

(2) Line 115, gene should be orthographic

Answer: Thanks for your kind comment. We have revised it in the new MS.

(3) Line 126, paired-end whole-genome sequencing should be deleted.

Answer: Thanks for your kind comment. We have revised it in the new MS.

(4) Line 170 hFL is hlf gene

Answer: Thanks for your kind comment. We have revised it in the new MS.

(5) in Table 2,there should have CYP81A6 analysis.

Answer: Thanks for your kind comment. The copy number of RNAi *CYP81A6* was analyzed with value of 2.00 (**Table 2**).

Reviewers' comments:

Reviewer #1 (Remarks to the Author):

Please see attached file

Reviewer #2 (Remarks to the Author):

The revised article has answered my question, and I think it meets the publication requirements.

General observation: The authors have taken their time to revise the manuscript and clarify some of the issues raised in the first round of review. Authors added another component where they put the *Gt1* gene sequences in the rice chromosome 5. They then did an analysis to prove that they can identify a cis gene in rice. Authors should highlight the assumptions of what they did which include: 1) During sequencing all the sequences will be as they envision 2) Only the inserted cis gene will be amplified during sequencing. The assumptions should be clearly highlighted in the manuscript. There are other corrections that need to be undertaken to enhance technical clarity of the manuscript. I have listed these per section below.

Novelty

The manuscript is not very high on novelty as similar reports using PE-WGS have been published. However, the integration of different approaches including digital drop-let PCR and the effort the authors took is commendable. Authors still face the challenges involved in picking up a cis gene in an endogenous genomic background. These challenges should be highlighted in the manuscript as actual challenges.

Title:

ok.

Abstract:

The abstract needs to be reviewed lines 18 – 22 still fronts the cisgenesis as the strength. Authors should check and have this rectified.

Line 18: Remove the words “novel” and “cisgenesis/intragenesis”

Materials and methods

Line 137: Did authors mean to use “...and...” instead of “...or...?”

Lines 144 – 146: Authors need to indicate the platforms they used for the different aspects of the analysis. They could even create a flow diagram on the same eg index removal, trimming etc

Lines 148 – 150: What was the set of paired end reads, which region was it corresponding to?

Line 149: Spiked into data?, which paired end reads?, basis for using

Line 172: Did the authors use DNA from entire plasmid or only the T-DNA region in filtering out?

Line 173-174: Authors should clarify how the native sequences were filtered, this is not clear as written.

Line 204/205: What the concentration of the 1 µl?

Line 233: Give the citation detailing the original TranSeq approach

Line 234: What was the basis for removing the false positives, what were these false positives?

Line 237: Change “a comparing...” with “A comparison...”

Lines 237 on false positives: The 253 false positives were based on the authors knowledge of the insertion location of the transgenic Rice line involved. The authors already knew the structure of the transgene in the genome, and the false positives were based on this

knowledge. If the authors didn't have this prior information, then these "false positives" would most likely be considered as part of the data.

Lines 236 – 243: The Gt1 promoter and *CYP81A6* gene were present in both plasmid, transgenic line and endogenously in the rice genome. Why would authors consider this as a false positive in the transgenic yet they are using controls where it is present? It can only be false positive if it was observed in the transgenic plant, yet it was not expected.

Lines 247 – 248: Authors have mentioned the 7 'sporadic' sequences were observed in the 62 unique sequences. Authors have referred these seven as "false positives", I am not sure these hits are unexpected. Since the sequencing approach used here was not targeting the transgene, then the argument that they are false positive may not be valid since other sequences in the genome would be expected. Authors should highlight this as a shortcoming to their approach and give possible solutions.

Line 250: Authors should strive to make their supplementary files comprehensible. As they are, a reader cannot tell apart the tables as the titles are truncated. Equally, I did not understand what they have pasted as supplementary file, I am only seeing a continuous string of characters. Authors should give meaning to the data by at least explaining what the files they have are. This should be done for all supplementary files they cited.

Lines 247-255: The PCRs of the false positive indicates that they were present in both negative control and transgenic. This therefore shows weakness in their approach,

Line 275: replace "nien" with "nine"

Line 277: replace "...including" with "included"

Line 519: "detail" to be replaced with "detailed"

Authors should also thoroughly go through the manuscript and correct other small errors and omissions that could not be listed here.

Figures and tables

Figure 2: It would be ideal if the authors have a section of how the reads mapped to the wildtype.

Figure 4: This figure does not have labels on which of the two panels is the WT and GM. Equally, authors mention that there were completely no sequences that were observed to hit the non-T-DNA region. However, figure 4 indicates some few alignments with the "Ori" region. Authors should explain this disparity. See below circled in red:

Supplemental Table S1: Seems to be truncated and incomprehensible. Is there a better way of summarizing this Table or presenting the data here?

Response to the comments

General observation: The authors have taken their time to revise the manuscript and clarify some of the issues raised in the first round of review. Authors added another component where they put the *Gt1* gene sequences in the rice chromosome 5. They then did an analysis to prove that they can identify a cis gene in rice. Authors should highlight the assumptions of what they did which include: 1) During sequencing all the sequences will be as they envision 2) Only the inserted cis gene will be amplified during sequencing. The assumptions should be clearly highlighted in the manuscript. There are other corrections that need to be undertaken to enhance technical clarity of the manuscript. I have listed these per section below.

Answer: Thanks for your affirmation on our revised MS. We have highlighted the assumptions of what we did on the analysis of simulated cisgenesis line in the new MS.

Novelty

The manuscript is not very high on novelty as similar reports using PE-WGS have been published. However, the integration of different approaches including digital droplet PCR and the effort the authors took is commendable. Authors still face the challenges involved in picking up a cis gene in an endogenous genomic background. These challenges should be highlighted in the manuscript as actual challenges.

Answer: Thanks for your comment. We have emphasized this point in the sections of introduction (Line 91-94; Line 109-111) and discussion (Line 4119-422) in the new MS.

Title:

ok.

Abstract:

The abstract needs to be reviewed lines 18 – 22 still fronts the cisgenesis as the strength. Authors should check and have this rectified.

Answer: Thanks for your comment. We have deleted the words of cisgenesis/intragenesis. (Line 18-19)

Line 18: Remove the words “novel” and “cisgenesis/intragenesis”

Answer: Thanks for your comment. We have removed these words. (Line 18-19)

Materials and methods

Line 137: Did authors mean to use” ...and...” instead of “...or...?”

Answer: Thanks for your comment. We have revised it. (Line 139)

Lines 144 – 146: Authors need to indicate the platforms they used for the different aspects of the analysis. They could even create a flow diagram on the same eg index removal, trimming etc.

Answer: Thanks for your comment. We have added the used tool of Trimmomatic in the MS. The quality control (QC) often includes the steps of removing adapters, low quality sequences, and contaminated sequences, which is common and basic step before further sequencing data analysis (Line 146-147). Therefore, we only showed this process with one general step of QC in Figure 1, and we added the annotation of QC in the figure legend. (Line 662-665)

Lines 148 – 150: What was the set of paired end reads, which region was it corresponding to?

Answer: Thanks for your comment. This paired end reads set was randomly generated from the mimic DNA containing three fragments of 1030 bp (rice Chr 05: 11030351-11031380), 2052 bp (Partial rice *Gt1* gene), and 1040 bp (rice Chr 05: 11031381-11032420). The sequences of the generated reads were shown in Supplemental Table S1. Also, we have rephrased it in the new MS. (Line 151-160)

Line 149: Spiked into data?, which paired end reads?, basis for using

Answer: Thanks for your comment. We have re-organized this paragraph for clear expressing the sequencing data of hypothetical *cisgenesis* line. The hypothetical *cisgenesis* line was designed and mimicked by inserting partial *Gt1* gene with length of 2052 bp into rice genome at the site of Chr 05 11031380. The mimic integration of *Gt1* gene has 4152 bp, containing three fragments of 1030 bp (rice Chr 05: 11030351-11031380), 2052 bp (Partial

rice *Gt1* gene), and 1040 bp (rice Chr 05: 11031381-11032420). Based on the sequence of mimic integration of *Gt1* gene (4152 bp), a set of paired-end reads was generated randomly generated *in silico* using art-illumina software with a sequencing depth of 29.28 × (As shown in **Supplemental Table S1**). The sequencing depth of generated read set is same with that of non-GM rice Xiushui line. Then, the raw data of hypothetical *cisgenesis* line was prepared by mixing the generated paired-end reads with the sequencing data of non-GM rice (Xiushui 110) for further analysis. Our design hopes to make the simulated data as consistent as possible with the real scene. (Line 151-160)

Line 172: Did the authors use DNA from entire plasmid or only the T-DNA region in filtering out?

Answer: Thanks for your comment. The entire plasmid was used in filtering out. We have added it in the new MS. (Line 175)

Line 173-174: Authors should clarify how the native sequences were filtered, this is not clear as written.

Answer: Thanks for your comment. We have described the steps to eliminate the reads from native sequence in Figure 1. Also, we explained each step in the revised MS. (Line 173-179)

Line 204/205: What the concentration of the 1 μl?

Answer: Thanks for your comment. The concentration of used template is 10 ng/μl. We have added the concentration in the new MS. (Line 210)

Line 233: Give the citation detailing the original TranSeq approach

Answer: Thanks for your comment. We have added the reference of “Yang et al., 2013” in the new MS. (Line 240)

Line 234: What was the basis for removing the false positives, what were these false positives?

Answer: Thanks for your comment. We have rephrased this sentence with “A total of 322

read pairs were extracted for which one end mapped to the rice reference genome and the other mapped to the transformed plasmid using the original TranSeq pipeline". The words of "without false-positive reads removed" were deleted. In this study, the false positives are the candidate reads from rice endogenous genome. (Line 239)

Line 237: Change "a comparing..." with "A comparison..."

Answer: Thanks for your comment. We have revised it. (Line 243)

Lines 237 on false positives: The 253 false positives were based on the authors knowledge of the insertion location of the transgenic Rice line involved. The authors already knew the structure of the transgene in the genome, and the false positives were based on this knowledge. If the authors didn't have this prior information, then these "false positives" would most likely be considered as part of the data.

Answer: Thanks for your comment. We do agree with your opinion. In this section, the word of "false positive" is not the most accurate term. We have deleted the words of "false positive" and rehearsed it in the whole MS. In original TranSeq analysis, much more candidate reads would be obtained including those from homologous sequence of endogenous genome, which significantly increased the difficulty in identifying the real and correct reads. However, those reads from homologous sequence of endogenous genome could be filtered out from the candidate reads with our modified TranSeq pipeline, which significantly narrows the candidate reads and improves the accuracy of candidate reads. In this study, we did not know the structure of the transgene integration in the genome of G281 line, and we only have the whole sequence of plasmid with the T-DNA. Through sequence BLASTN analysis, we knew that the Gt1 promoter and *CYP81A6* gene in the T-DNA region of plasmid was from rice genome. Then, we can filter out the reads generated from *Gt1* promoter and *CYP81A6* gene using modified TranSeq pipeline, and obtain the more accurate candidate reads. (Line 244-248)

Lines 236 – 243: The Gt1 promoter and *CYP81A6* gene were present in both plasmid, transgenic line and endogenously in the rice genome. Why would authors consider this as a false positive in the transgenic yet they are using controls where it is present? It can only be

false positive if it was observed in the transgenic plant, yet it was not expected.

Answer: Thanks for your comment. In this section, the word of “false positive” is not the most accurate term. We have deleted the words of “false positive” and rephrased it in the whole MS. Please find it as the above question discussed. (Line 240-248; Line 281-286)

Lines 247 – 248: Authors have mentioned the 7 ‘sporadic’ sequences were observed in the 62 unique sequences. Authors have referred these seven as “false positives”, I am not sure these hits are unexpected. Since the sequencing approach used here was not targeting the transgene, then the argument that they are false positive may not be valid since other sequences in the genome would be expected. Authors should highlight this as a shortcoming to their approach and give possible solutions.

Answer: Thanks for your comment. In this section, the word of “false positive” is not the most accurate term. We have deleted the words of “false positive” and rephrased it with the sentence of “The other seven sporadic read pairs mapped to other chromosomes were also confirmed to be untrue according to BLASTN analysis and PCR amplification.” in the whole MS. (Line 252-254) Although, the sporadic sequence could be often generated in whole genome resequencing analysis, we could use various method to validate these reads, such as BLASTN, IGV view, and PCR amplification. We have highlighted these methods in the revised MS. (Line 260-263)

Line 250: Authors should strive to make their supplementary files comprehensible. As they are, a reader cannot tell apart the tables as the titles are truncated. Equally, I did not understand what they have pasted as supplementary file, I am only seeing a continuous string of characters. Authors should give meaning to the data by at least explaining what the files they have are. This should be done for all supplementary files they cited.

Answer: Thanks for your comment. Originally, we combined all the supplemental tables in one excel file. In the new version, we have separated the tables in separate excel files.

Lines 247-255: The PCRs of the false positive indicates that they were present in both negative control and transgenic. This therefore shows weakness in their approach,

Answer: Thanks for your comment. In the whole genome resequencing analysis, sporadic

reads are often existed because of the sequencing error, accidental result, noise sequences, and alignment jitter. This is the limitation of the whole genome re-sequencing analysis. However, the number of sporadic reads is so far below the number of sequenced that it can be considered negligible. We have pointed out this limitation in the revised MS. (Line 260-263)

Line 275: replace “nien” with “nine”

Answer: Thanks for your comment. We have revised it. (Line 284)

Line 277: replace “...including” with “included”

Answer: Thanks for your comment. We have revised it. (Line 287)

Line 519: “detail” to be replaced with “detailed”

Answer: Thanks for your comment. We have revised it. (Line 532)

Authors should also thoroughly go through the manuscript and correct other small errors and omissions that could not be listed here.

Answer: Thanks for your comment. We have read and check the whole MS to correct the typing errors and omissions. All revised place was highlighted with red color.

Figures and tables

Figure 2: It would be ideal if the authors have a section of how the reads mapped to the wildtype.

Answer: Thanks for your comment. The Figure 2A is performed using the 69 candidate read pairs mapped to rice reference genome sequence (GCA_001433935.1) by clustering pattern analysis. We have modified the figure with additional annotation, and added one brief introduction of how the reads mapped to the rice reference genome sequence in the revised MS. (Line 264-267)

Figure 4: This figure does not have labels on which of the two panels is the WT and GM. Equally, authors mention that there were completely no sequences that were observed to

hit the non-T-DNA region. However, figure 4 indicates some few alignments with the “Ori” region. Authors should explain this disparity. See below circled in red:

Answer: Thanks for your comment. We found that there are four reads and two reads mapped to the Ori region in WT and G281, respectively. However, these sequences might come from the rice genome background or the sequencing contamination considering the sequencing depth of ~29 X. Therefore, we believed that these sequences were not from the plasmid backbone really. Also, we rephrased and discussed the results of IGV analysis in the new MS. (Line 340-348)

Supplemental Table S1: Seems to be truncated and incomprehensible. Is there a better way of summarizing this Table or presenting the data here?

Answer: Thanks for your comment. We have deleted the Information that was not particularly important from this table, and only keep the name and sequence of the generated read pairs.

REVIEWERS' COMMENTS:

Reviewer #1 (Remarks to the Author):

I commend the efforts taken by the authors to improve the manuscript based on previous comments. The paper is now clear to me and I approve its publication. I have no additional comments.